# Synthetic GPR84 Agonists in Colorectal Cancer: Effective in THP-1 Cells but Ineffective in BMDMs and MC38 Mouse Tumor Models

**DOI:** 10.3390/ijms26020490

**Published:** 2025-01-09

**Authors:** Marlene Schwarzfischer, Maria Rae Walker, Michele Curcio, Nader M. Boshta, Arnaud Marchand, Erik Soons, Doris Pöhlmann, Marcin Wawrzyniak, Yasser Morsy, Silvia Lang, Marianne Rebecca Spalinger, Matthias Versele, Michael Scharl

**Affiliations:** 1Department of Gastroenterology and Hepatology, University Hospital Zürich, University of Zürich, 8091 Zürich, Switzerland; marlene.schwarzfischer@usz.ch (M.S.); maria.walker@usz.ch (M.R.W.); doris.poehlmann@usz.ch (D.P.); marcin.wawrzyniak@usz.ch (M.W.); yasser.morsy@usz.ch (Y.M.); silvia.lang@usz.ch (S.L.); marianne.spalinger@usz.ch (M.R.S.); 2CISTIM Leuven vzw, Gaston Geenslaan 2, 3001 Leuven, Belgium; michele.curcio@cistim.be (M.C.); nader.boshta@cistim.be (N.M.B.); arnaud.marchand@cistim.be (A.M.); erik.soons@cistim.be (E.S.); matthias.versele@cistim.be (M.V.); 3Centre for Drug Design and Discovery (CD3), KU Leuven, Gaston Geenslaan 2, 3001 Leuven, Belgium

**Keywords:** colorectal cancer (CRC), tumor-associated macrophages (TAMs), GPR84, macrophage activation, 6-OAU, ZQ-16, anti-cancer therapy, immunotherapy

## Abstract

Tumor-associated macrophages (TAMs) in the colorectal cancer (CRC) microenvironment promote tumor progression but can be reprogrammed into a pro-inflammatory state with anti-cancer properties. Activation of the G protein-coupled receptor 84 (GPR84) is associated with pro-inflammatory macrophage polarization, making it a potential target for CRC therapy. This study evaluates the effects of the GPR84 agonists 6-OAU and ZQ-16 on macrophage activation and anti-cancer efficacy. GPR84 expression on THP-1 macrophages and murine BMDMs was analyzed using flow cytometry. Macrophages were treated with 6-OAU or ZQ-16, and pro-inflammatory cytokine levels, reactive oxygen species (ROS) production, and phagocytosis were assessed using qPCR and functional assays. Anti-cancer effects were tested in a subcutaneous MC38 tumor model, with oral or intraperitoneal agonist administration. Pharmacokinetics and compound stability were also evaluated. In THP-1 macrophages, 6-OAU increased pro-inflammatory cytokines and ROS production, with ZQ-16 showing similar effects. However, neither agonist induced pro-inflammatory responses, ROS production, or phagocytosis in murine macrophages. In vivo, both agonists failed to inhibit tumor growth in the MC38 model despite systemic exposure. Current GPR84 agonists lack efficacy in promoting anti-cancer macrophage activity, limiting their potential as CRC therapies.

## 1. Introduction

Colorectal cancer (CRC) is the third-most frequent and second-most lethal cancer worldwide, with poor prognosis for metastatic cases due to limited treatment options [1,2,3,4]. Despite advancements like immune checkpoint inhibitors, their efficacy in CRC is insufficient, highlighting the urgent need for novel therapies.

Macrophages have gained attention as potential targets for CRC therapy [5,6] due to their ability to polarize into pro- or anti-inflammatory subtypes [7]. Tumor-associated macrophages (TAMs) in the CRC tumor microenvironment [8] promote progression and metastasis by enhancing angiogenesis, extracellular matrix remodeling, and tumor cell proliferation [5,8]. Reprogramming TAMs offers a promising therapeutic strategy [8,9,10].

Macrophage-mediated cytotoxicity, phagocytosis, and tumor microenvironment reshaping are key goals for novel CRC therapies. GPR84, a G protein-coupled receptor (GPCR) primarily expressed on immune cells [11], may be a promising target for cancer treatment. It promotes pro-inflammatory macrophages in inflammatory conditions [12,13,14], with increased expression in inflamed tissues [11] and activation of the receptor was shown to enhance inflammatory processes in experimental animal models of endotoxemia and multiple sclerosis [15], fibrosis [16], and pulmonary inflammation [17]. Moreover, in mouse models of neuropathic pain [14], non-alcoholic steatohepatitis [12], and colitis [13], GPR84 deletion or pharmacological inhibition correlated with the reduction of pro-inflammatory macrophages and decreased disease severity.

GPR84 plays a key role in cell signaling and is believed to sense medium-chain fatty acids (MCFAs) [18,19] and potentially bacterial metabolites in inflammatory conditions [20]. Though its structure is well characterized [19,21], GPR84 is still considered an orphan receptor due to limited data on its binding interactions [22,23]. GPCRs like GPR84 can trigger various signaling pathways depending on the agonist, a phenomenon known as signaling bias [24]. Recently, biased GPR84 agonists [22] like 6-OAU [25] and ZQ-16 [26] were developed, showing potent activation of GPR84 and promoting pro-inflammatory responses in macrophages [11,13,25,27,28,29], such as cytokine production, migration, and phagocytosis of cancer cells. These findings suggest potential therapeutic applications in cancer treatment.

This study aims to investigate the effects of GPR84 agonists (6-OAU and ZQ-16) on macrophage activation in human THP-1 cells and murine bone marrow-derived macrophages (BMDMs). Specifically, we evaluate the agonists’ impact on pro-inflammatory cytokine production, reactive oxygen species (ROS) generation, and phagocytosis. By comparing responses in THP-1 and BMDM macrophages, this study seeks to uncover species-specific differences and gain insights into the potential therapeutic applications of GPR84 activation in modulating immune responses. Additionally, this study includes in vivo experiments using the MC38 subcutaneous tumor model to assess the anti-cancer potential of these agonists and their effects on tumor growth.

## 2. Results

### 2.1. 6-OAU Triggered the Production of Pro-Inflammatory Cytokines and ROS in Differentiated THP-1 Cells

First, we verified the surface expression of GPR84 on human THP-1 macrophages by using flow cytometry. Flow cytometry gating is shown in Appendix A, and data on antibody specificity are given in Appendix A. These analyses revealed that GPR84 is detectable on the surface of PMA-differentiated THP-1 cells and that surface and mRNA expression levels of the receptor were significantly increased upon pro-inflammatory stimulation with LPS (Figure 1A). The macrophages were then incubated with the synthetic GPR84 agonist, 6-OAU, at a concentration chosen based on intracellular cAMP measurements previously published by Recio et al. [11]. In this setting, 6-OAU increased mRNA expression of the pro-inflammatory cytokines interleukin (IL) 6, IL12b, and tumor necrosis factor α (TNFa) in lipopolysaccharide (LPS) pre-treated THP-1 cells (Figure 1B). Genes of interests were chosen based on data published by Recio et al. [11]. Furthermore, CC-chemokin-ligand-2 (CCL2) mRNA expression was increased upon 6-OAU incubation without LPS priming (Figure 1B). Information on the synthesis and quality control of 6-OAU is given in Appendix A. The activity of 6-OAU with human GPR84 was confirmed in a functional assay based on measurements of cyclic adenosine 3′,5′-monophosphate (cAMP) (Appendix A). Data on the protein plasma binding (PPB) and metabolic stability of 6-OAU are given in Appendix A. Moreover, the production of ROS was significantly elevated in differentiated THP-1 cells stimulated with 6-OAU (Figure 1C) and inhibited in the presence of a GPR84 antagonist (Appendix A). Contrary to previous publications [11], the phagocytotic efficacy of fluorescent polystyrene (PS) beads was not increased in differentiated THP-1 cells upon 6-OAU incubation (Figure 1D) regardless of whether the cells were pre-treated with LPS treatment or not. Gating of the flow cytometry analysis and the respective controls are shown in Appendix A. To summarize, these findings indicate that 6-OAU triggered pro-inflammatory signaling cascades and oxidative bursts in THP-1 macrophages.

### 2.2. 6-OAU Did Not Enhance Pro-Inflammatory Signaling nor Phagocytosis in Murine BMDMs

Since immortalized cell lines such as THP-1 often do not feature canonical functions of their in vivo counterparts, we aimed to test the effects of GRP84 agonists in primary macrophages. In view of the mouse cancer models performed later in this study, we explored the effects of the GPR84 agonist 6-OAU on murine BMDMs. 

In accordance with the data on human THP-1 cells, LPS treatment significantly increased GPR84 surface levels and mRNA expression in BMDMs (Figure 2A). Flow cytometry gating is shown in Appendix A, and data on antibody specificity are given in Appendix A. However, in contrast to the effects observed in THP-1 cells, 6-OAU did not elevate the mRNA expression of the assessed pro-inflammatory cytokines (Figure 2B). Furthermore, the phagocytotic efficacy of BMDMs was not enhanced upon 6-OAU stimulation, as demonstrated in a phagocytosis assay using PS beads (Figure 2C), irrespective of LPS pre-treatment of the cells. Gating of the flow cytometry analysis is shown in Appendix A. Similar results were obtained in an antibody-dependent cell phagocytosis (ADCP) assay using fluorescently labeled MC38 murine colorectal cancer cells which were pre-incubated with an anti-CD47 antibody to induce phagocytosis (Figure 2D). Binding to CD47 was shown to block the “don’t eat me” signal on the surface of the target cells and hence induced macrophage-mediated phagocytosis [29]. However, antibody-dependent phagocytosis of cancer cells was not enhanced by GPR84 agonism. Gating of the flow cytometry analysis, the respective staining controls, and CD47 surface expression on MC38 cells are shown in Appendix A. Collectively, these data indicate that 6-OAU was inducing neither a pro-inflammatory response nor phagocytotic activities in murine BMDMs, despite the high surface expression of GPR84.

### 2.3. ZQ-16 Did Not Induce a Pro-Inflammatory Response in Murine BMDMs

Since 6-OAU failed to activate a pro-inflammatory response in murine BMDMs, we tested the more potent GPR84 agonist, ZQ-16, as measured in a functional assay based on measurements of cAMP (Appendix A). Data on the PPB and metabolic stability of ZQ-16 are given in Appendix A. Information on the synthesis and quality control of ZQ-16 is given in Appendix A. Data on the specificity and safety of ZQ-16 are given in Appendix A and Appendix A, respectively. In BMDMs, ZQ-16 failed to increase mRNA expression levels of the pro-inflammatory cytokines tested (Figure 3A) regardless of LPS pre-treatment of the cells. Furthermore, ZQ-16 did not induce oxidative bursts (Figure 3B) or phagocytosis in LPS pre-treated BMDMs. However, in THP-1 cells, ZQ-16 administration induced the production of ROS, which was inhibited in the presence of a GPR84 antagonist (Appendix A). EC50 data comparing the potency of both GPR84 agonists are provided in Appendix A. Collectively, these data indicate that despite the high specificity and binding affinity to the GPR84 receptor, ZQ-16 was neither able to promote the production of pro-inflammatory cytokines or ROS, nor to increase phagocytotic efficacy in murine BMDMs.

### 2.4. Neither 6-OAU nor ZQ-16 Enhanced Tumor Cell Phagocytosis in Macrophages

Previous publications reported that GPR84 activation promotes macrophage-mediated tumor cell phagocytosis [13]. To test the effect of 6-OAU and ZQ-16 on the efficacy of tumor cell phagocytosis, a further ADCP assay was performed. pHrodo Red-labeled human Raji cancer cells (target cells) were incubated with magrolimab (anti-CD47 antibody) and co-cultured with calcein AM-labeled J774 mouse macrophages which were pre-treated with LPS and stimulated with either 6-OAU or ZQ-16. LPS pre-incubation of macrophages significantly increased Raji cell phagocytosis in J774 cells, and the effect was enhanced by the addition of magrolimab (Appendix A). However, both GPR84 agonists failed to increase the phagocytosis of Raji cells by means of J774 macrophages (Figure 4). Conversely, 6-OAU stimulation of J774 cells significantly decreased the phagocytosis of Raji cells (Figure 4A). In summary, the ADCP assay indicated that the GPR84 agonists 6-OAU and ZQ-16 did not increase macrophage-mediated tumor cell phagocytosis.

### 2.5. 6-OAU Had No Anti-Cancer Effect in the Subcutaneous MC38 Tumor Injection Model

The immunological function of GPR84 is poorly studied. Hence, it is possible that 6-OAU activates other immune cell types with anti-cancer capacities or that complex immune cell interactions are required to exert macrophage-mediated anti-cancer effects. Therefore, we tested the anti-cancer potential of 6-OAU in vivo using the MC38 tumor injection model. The mice were treated with the vehicle or 1 or 10 mpk 6-OAU from day 6 to day 14 after tumor cell injection via oral gavage. The experimental design is shown in Figure 5A. 6-OAU treatment failed to reduce tumor growth in this model (Figure 5B) and no significant difference was observed when comparing the average tumor growth between treatment groups (Figure 5C). Mice were euthanized as soon as the termination criteria (stated in the Materials and Methods Section) were reached. The removal of single mice from the experiment is indicated with crosses on the average tumor growth curve and displayed on the survival curve in Figure 5C. 6-OAU did not reduce the terminal tumor volume nor terminal tumor weight (Figure 5D). Pharmacokinetic data of 6-OAU are shown in Appendix A, as well as in Appendix A. In conclusion, despite the systemic uptake of the agonist, 6-OAU had no anti-cancer effect in the subcutaneous MC38 tumor injection model.

### 2.6. ZQ-16 Had No Anti-Cancer Effect in the Subcutaneous MC38 Tumor Injection Model

To evaluate the effect of ZQ-16 on the complexity of the entire immune system, its anti-cancer potential was also evaluated in the subcutaneous MC38 tumor cell injection model. Considering the lack of efficacy with 6-OAU after 1 week of dosing, we decided to dose the animals with ZQ-16 for 2 weeks using two different doses and administration routes. From day 7 to 21 after tumor cell injection, the mice were treated either with the vehicle or 0.457 mpk ZQ-16 applied via intraperitoneal injections or with the vehicle or 10 mpk ZQ-16 applied via oral gavage. The experimental design is shown in Figure 6A. ZQ-16 did not reduce tumor growth (Figure 6B) and no significant difference was observed when comparing the average tumor growth between treatment groups (Figure 6C). The removal of single mice from the experiment is indicated with crosses on the average tumor growth curve and displayed on the survival curve in Figure 6C. The termination criteria are stated in the Materials and Methods section. ZQ-16 did not reduce the terminal tumor volume nor the terminal tumor weight (Figure 6D). The activity of ZQ-16 on the mouse isoform of GPR84 was tested in a cAMP signaling assay (Appendix A). Pharmacokinetic data of ZQ-16 are shown in Appendix A, as well as in Appendix A. In summary, despite the systemic uptake of ZQ-16, the GPR84 agonist was not able not reduce tumor growth in the subcutaneous MC38 tumor injection model.

## 3. Discussion

Synthetic orthosteric GPR84 agonists were previously shown to induce pro-inflammatory signaling cascades and ROS production, as well as enhance phagocytotic capacity in macrophages [30]. These functions crucially contribute to macrophage-mediated cytotoxic killing and clearance of cancer cells. Thus, we hypothesized that activation of GPR84 by synthetic agonists might be a powerful new approach for the treatment of CRC. In the present study, we demonstrate that 6-OAU triggered pro-inflammatory effects in human THP-1 macrophages, while both GPR84 agonists 6-OAU and ZQ-16 failed to induce pro-inflammatory responses in murine BMDMs. Most importantly, both agonists failed to enhance macrophage-mediated phagocytosis of cancer cells in vitro and to reduce tumor growth in the subcutaneous MC38 tumor cell injection model in vivo. However, to assess the clinical potential of GPR84 agonists for CRC treatment, additional experiments are necessary.

Mouse and human GPR84 share 85% identity in their extracellular regions and transmembrane domains [31]. However, data comparing the affinity of synthetic agonists to murine and human GPR84 receptors are currently not available. Yet, we and others have previously shown that GPR84 agonists activate both human and murine GPR84. Nevertheless, subtle differences in the structure could result in differences in receptor affinity with synthetic agonists, which could explain the differences between murine BMDMs and human THP-1 cells with regards to inflammatory cytokine expression upon 6-OAU stimulation. Additionally, while murine BMDMs closely recapitulate the functions and properties of in vivo monocyte-derived macrophages, immortalized cell lines such as THP-1 may differ from primary macrophages with respect to their phagocytic activity, cytokine production, and regulation of oxidative bursts [32]. 

Our findings seem contradictory to previous publications [11,19,29]. Recio et al. reported enhanced mRNA expression of the pro-inflammatory cytokines Tnfa, Il6, Il12, and Ccl2 and enhanced phagocytosis of opsonized PS beads in LPS-treated murine BMDMs upon 6-OAU stimulation [11]. While the experimental setup was comparable in both studies, the effects described by Recio et al. could not be reproduced. Contrary results may be explained by the differences in the generation and characteristics of murine BMDMs and LPS pre-incubation times. Kamber et al. described increased phagocytosis of human RAMOS cancer cells after CD47 blockage with an anti-CD47 antibody (B6H12) in J774 mouse macrophages upon 6-OAU and ZQ-16 stimulation [29]. This study was supported by Zhang et al. who reported an increased phagocytotic rate of human Raji cancer cells treated with an anti-CD47 antibody (B6H12) in murine BMDMs (generated from Balb/c mice) that were stimulated with 6-OAU [13]. These results could not be validated in our study where an analogous ADCP assay was performed using a J774 effector and Raji target cells incubated with a different anti-CD47 antibody (Magrolimab). Discrepancies between the studies might result from the different target cells used and the species they were derived from. Supportively, 6-OAU failed to induce a pro-inflammatory response in LPS pre-treated microglia, the myeloid cells of the central nervous system [33], indicating that GPR84 signaling may trigger different downstream effects in myeloid cells originated from different tissues. Luscombe et al. recently expressed concerns claiming that the variety of cell types and stimulation paradigms do not allow for a comparison between GPR84 studies nor to draw conclusions about agonist-specific functions [30]. Therefore, the authors postulated three key factors that should be considered in order to study the effects of GPR84 activation in vitro: (1) the stimulation of GPR84 surface expression is required depending on the cell type, (2) cell priming is required for certain assays, and (3) an initial response is required to see GPR84-augmented effects [30]. All of these factors contribute to the differences between our study and previous reports. While the present study mainly focused on GPR84-mediated effector functions in macrophages, one should keep in mind that GPR84 activation has been associated with chemotaxis and oxidative bursts in neutrophils [17], which may constitute another interesting target for CRC therapy. 

Recently, Li et al. reported that GPR84 serves as an important metabolic sensor for orchestrating TAM polarization and initiating an anti-tumor response [34]. According to the authors, i.p. injection of 6-OAU (250 µg from day 8 to 14 consecutively once per day) significantly delayed tumor growth in the subcutaneous MC38 tumor cell injection model [34]. In contrast, in the present study, we show that both 6-OAU and ZQ-16 failed to reduce tumor growth in the same model. Although the 6-OAU concentrations used in both studies (250 µg 6-OAU in a 25 g mouse corresponds to 10 mpk) are comparable, the studies face disparities regarding the dosing frequency, formulation, and administration route of 6-OAU, which might account for the different study outcomes. Interestingly, GPR84 has been associated with the sensing of bacterial products and metabolites [20,35]. In this context, Peters et al. suggested that GPR84 signaling might enable differentiation between beneficial bacterial metabolites from commensal bacteria and those from potentially harmful pathogens [30,35]. Therefore, the question arises as to whether the observed discrepancies might possibly be a result of deviations in the microbiome of the animal husbandries that affect the outcome of the two studies. The variance in the microbiome between experimental animal facilities is a major threat to reproducibility and translatability in research [36]. Interestingly, commensal bacteria are found to affect carcinogenesis, CRC formation, and progression and most importantly to modulate anti-tumor immune response and therapeutic success [37]. Hence, the over-representation of beneficial commensals in an animal husbandry might innately induce an anti-tumor response via GPR84 activation, obscuring any add-on effects using activating agonists. Although GPR84 is currently considered an orphan receptor, the identification of more potent natural agonists, such as bacteria-derived medium-chain fatty acids, might be of great interest with regard to novel bacteria- or metabolite-based CRC therapeutic approaches. 

Despite the negative data presented in the current study, GPR84 remains an interesting candidate to be explored in the context of CRC and other cancer entities. Recently, Jian et al. reported that GPR84 activation prevents osteolysis during bone metastases of CRC and that CRC cells cause downregulation of GPR84 expression on BMDMs, promoting osteoclastogenesis. The authors concluded that GPR84 might be a potential therapeutic target to prevent bone destruction by CRC metastasis [38]. Additionally, GPR84 was found to be negatively correlated with the prognosis of esophageal cancer and to be overexpressed on myeloid-derived suppressor cells, which were discovered to accumulate in this disease. Quin et al. demonstrated that the blockade of GPR84 inhibited the tumor progression of orthotropic esophageal cancer mouse models as well as subcutaneous LLC and B16 models by remodeling the immunosuppressive tumor microenvironment. Specifically, the GPR84 blockade decreased the immunosuppressive activity of myeloid-derived suppressor cells and resulted in the recruitment and activation of cytotoxic CD8+ T-cells. Consequently, GPR84 antagonism significantly enhanced the therapeutic success of anti PD-1 immunotherapy in the LLC/B16 esophageal cancer model [39]. Follow-up studies are required to evaluate whether the proposed mechanism can be translated to other cancer entities. 

The in vivo studies were limited by the use of a single mouse model, a suboptimal dosing regimen, and only two specific GPR84 agonists, suggesting the need for more physiologically relevant models, optimized dosing strategies, and a broader range of agonists to fully evaluate the potential of GPR84 activation in CRC therapy. Although the subcutaneous MC38 tumor cell injection model is frequently used for preclinical drug screening approaches, it is considered rather artificial as it is lacking the physiological tumor environment and the proximity to the intestinal microbiome [40]. Therefore, orthotopic mouse models of CRC might be more adequate and informative to evaluate the anti-cancer potential of synthetic GPR84 agonists. Further, mouse models featuring other cancer entities could help to evaluate the anti-cancer effect of synthetic GPR84 agonists. Although different application methods and doses were selected based on the plasma availability determined in PK studies, previous publications indicate internalization of the receptor upon agonist stimulation especially with 6-OAU [41]. The establishment of an optimal dosing regimen, which simultaneously enables activation of the GPR84 receptor while preventing internalization and thereby maintaining responsiveness, poses a challenge for future studies. Signaling bias of synthetic GPR84 agonists offers great opportunities to increase specificity and potency while reducing unwanted off-target effects. However, differently biased GPR84 agonists result in the activation of distinct signaling cascades initiating distinct cellular responses [24,30]. Thus, it is certainly possible that the choice of agonists used in the current study might have been suboptimal to test our hypothesis. Thorough investigation of the effects on physiological macrophage functions caused by biased GPR84 signaling represents a future challenge that must be overcome to draw a general conclusion about the potential of GPR84 agonists for the development of novel therapeutic strategies in CRC.

In conclusion, while our findings do not support the therapeutic efficacy of synthetic GPR84 agonists for CRC at this stage, they contribute valuable negative data, which are often underappreciated in scientific progress. The inability to replicate previous results underscores important gaps in our understanding of GPR84’s role in cancer therapy and highlights the need for careful refinement of experimental designs, agonist selection, dosing regimens, and model systems. Our study also emphasizes the importance of investigating species- and cell-type-specific responses to GPR84 activation. While our study does not fully substantiate GPR84 as a significant therapeutic target in CRC, it provides important insights into the complexities and potential limitations of targeting GPR84 for treatment. As suggested, exploring diverse GPR84-targeting strategies, including biased agonism, alternative receptor pathways, and the microbiome’s role in modulating GPR84 activity, is crucial for advancing our understanding of its therapeutic potential.

## 4. Materials and Methods

### 4.1. Cultivation, Differentiation, and Stimulation of THP-1 Cells

Human THP-1 monocytes (ACC 16, DMSZ) were expanded in 150 cm^2^ cell culture flasks (TPP) in 25 mL RPMI 1640 media (Thermo Fisher Scientific, Waltham, MA, USA), supplemented with 10% FCS (#S181T-500, VWR, Radnor, PA, USA). For macrophage differentiation, cells were counted, centrifuged at 1700 rpm for 5 min, and re-suspended in RPMI 1640 media supplemented with 10% FCS and 12.5 ng/mL Phorbol 12-myristate 13-acetate (PMA) (Sigma Aldrich, St. Louis, MO, USA) in a ratio of 1 × 10^6^ cells per mL. Subsequently, the cells were seeded in tissue-culture treated plates (TPPs) (6-well plate: 2 × 10^6^ cells in 2 mL media; 12-well plate: 1 × 10^6^ cells in 1 mL media and 0.5 × 10^6^ cells in 0.5 mL media; 96-well plate: 0.1 × 10^6^ cells in 0.1 mL media). In vitro assays were performed once the cells were attached to the plate. For experiments, the macrophages were incubated ± LPS (InvivoGen, San Diego, CA, USA), reconstituted in ddH_2_O, and diluted at different concentrations in RPMI 1640 + 2% FCS for the indicated time period at 37 °C with 5% CO_2_. 

### 4.2. Isolation, Cultivation, and Stimulation of Bone Marrow-Derived Macrophages

Bone marrow was isolated from 8–12-week-old mice, homogenized with an 18-gauge needle, and passed through a 70 µm cell strainer, as previously described [42]. Cells were seeded in non-tissue culture-treated polystyrene 94 × 6 mm Petri dishes (Greiner Bio-One) at a density of approximately 3 × 10^6^ cells in 10 mL RPMI 1640 media, supplemented with 10% FCS, 1 mM Sodium Pyruvate (Thermo Fisher Scientific), 1 mM L-Glutamine (Thermo Fisher Scientific), 50 units/mL Penicillin 50 µm/mL Streptomycin (Thermo Fisher Scientific), and 10% in-house produced MCSF, and cultivated at 37 °C with 10% CO_2_. The production and testing of in-house MCSF is described in the Appendix A. The media was topped up on day 3 (5 mL per Petri dish) and completely replaced on day 5. On day 7, the cells were washed with PBS (5 mL, pH 7.4) and detached from the Petri dishes using 2 mL cold Accutase (Chemie Brunschwig, Basel, Switzerland) on ice for 10 min. The reaction was stopped by adding RPMI + 10% FCS (3× volume of Accutase). The cells were counted, centrifuged at 1700 rpm for 5 min, and re-suspended in RPMI 1640 media supplemented with 10% FCS. For the experiments, BMDMs were seeded in non-tissue culture-treated plates (Starlab Schweiz AG, Affoltern am Albis, Switzerland) (6-well plate: 2 × 10^6^ cells in 2 mL media; 12-well plate: 1 × 10^6^ cells in 1 mL media; 24-well plate: 0.5 × 10^6^ cells in 0.5 mL media). In vitro assays were performed once the cells were attached to the plate. For the experiments, the macrophages were incubated ± LPS (InvivoGen), reconstituted in ddH_2_O, and diluted at different concentrations in RPMI 1640 + 2% FCS for the indicated time period at 37 °C with 10% CO_2_.

### 4.3. Quantification of GPR84 Surface Expression

Differentiated human THP-1 cells and murine BMDMs were stimulated ± 100 ng/mL LPS (InvivoGen) for 16 h in RPMI + 2% FCS (12-well plate: 1 × 10^6^ cells in 1 mL media). Subsequently, the media was removed and the cells were washed three times with PBS (1 mL, pH 7.4) and detached with 5 mM EDTA in PBS (1 mL, pH 7.4) on ice for 10 min. The cells were collected in FACS tubes, the reaction was stopped with RPMI 1640 + 10% FCS (3× the volume of EDTA/PBS), and the cells were centrifuged at 1700 rpm for 5 min. The pellets were washed twice with PBS (0.5 mL, pH 7.4), and incubated with an anti-GPR84 antibody (rabbit pAb, #DF2769-200UL, Affinity Bioscience, Cincinnati, OH, USA) diluted at 1:100 in PBS (0.1 mL, pH 7.4) for 1 h at 4 °C. The cells incubated with the secondary antibody only were used as the control. The cells were centrifuged at 1700 rpm for 5 min, washed twice with PBS (0.5 mL, pH 7.4), and incubated with an anti-rabbit AlexaFluor594 conjugated secondary antibody (goat pAb, #A11081, Thermo Fisher Scientific) diluted at 1:1000 and with a Zombie NIRTM Fixable Viability Kit (#423106, BioLegend, San Diego, CA, USA) diluted at 1:500 in PBS (0.1 mL, pH 7.4) for 1 h. The cells were centrifuged at 1700 rpm for 5 min, washed twice with PBS (0.5 mL, pH 7.4), and fixed in 0.1% PFA/PBS (0.2 mL, pH 7.4). Samples were acquired using an LSR II Fortessa cytometer equipped with 405 nm, 488 nm, 561 nm, and 640 nm laser lines (BD Bioscience, San Jose, CA, USA) and FACSDiva Software v9.0 (BD Bioscience). GPR84 surface expression was quantified by manual gating using FlowJo v10.10 (BD Bioscience).

### 4.4. Agonist Preparation for In Vitro Experiments

The GPR84 agonists 6-OAU and ZQ-16 were synthetized and provided by CISTIM, Leuven, Belgium. For in vitro experiments, the compounds were initially dissolved in 100% dimethyl sulfoxide (DMSO) (Sigma Aldrich) and sonicated for 1 min to retrieve a 50 mM stock solution. The stock solution was further diluted with the respective assay buffer to obtain the desired concentration for the in vitro experiments. Assay buffer with an equal amount of DMSO was used as a control for each experiment.

### 4.5. Quantification of mRNA Levels

Differentiated human THP-1 cells and murine BMDMs seeded in 24-well plates (0.5 × 10^6^ cells in 0.5 mL media) were stimulated ± LPS (InvivoGen) for 16 h at 37 °C with 5% CO_2_ followed by incubation with 6-OAU or ZQ-16 (incubation times and concentrations are stated in the respective figure legends). Agonists were removed and the cells were washed twice with PBS (0.5 mL, pH 7.4). RNA was isolated using a Maxwell RSC simplyRNA Cell kit and the Maxwell RSC instrument (Promega, Madison, WI, USA) according to the manufacturer’s instructions. RNA concentration was quantified, measuring absorbance at 260 nm and 280 nm, using a Synergy H1 microplate reader (BioTek, Winooski, VT, USA). Complementary DNA (cDNA) transcription was performed using an Applied BiosytsemsTM High-Capacity cDNA Reverse Transcription Kit (Thermo Fisher Scientific). Gene expression was analyzed using pre-designed TaqMan Real-Time PCR Assays and TaqMan FAST Universal PCR Master Mix on a QuantStudio 6 Flex Real-Time PCR System according to the manufacturer’s instruction (all components purchased from Thermo Fisher Scientific). Measurements were performed in triplicates, and the results were normalized to *Actb* (Thermo Fisher Scientific) as the endogenous control and analyzed according to the ∆∆CT method. THP-1 human TaqMan assays: IL6 Hs00174131_m1, IL12b Hs01011518_m1, CCL2 Hs00234140_m1, TNFa Hs00174128_m1, GPR84 Hs00220561_m1, and Human ACTB (Beta Actin) Endogenous Control (VIC^®^/MGB Probe, Primer Limited) #4326315E. BMDMs murine TaqMan assays: Il6 Mm00446190_m1, Il12b Mm00434174_m1, Ccl2 Mm00441242_m1, Tnfa Mm99999068_m1, Gpr84 Mm00518921_m1, and mouse *Actb* (actin, beta) Endogenous Control (VIC^®^/MGB Probe, Primer Limited) #4352341E.

### 4.6. Bead Phagocytosis Assay

Differentiated human THP-1 cells and murine BMDMs seeded in 24-well plates were stimulated ± 100 ng/mL LPS (InvivoGen) for 16 h followed by incubation with 6-OAU or ZQ-16 (incubation times and concentrations are stated in the respective figure legends). Meanwhile, 3 µm polystyrene (PS) particles (#PS-FluoGrün-3.0, *w*/*v* 2.5%, micro particles GmbH, Berlin, Germany) were opsonized with human serum (Sigma Aldrich) diluted in bead solution at 1:10. Subsequently, the agonist was removed, and the cells were washed twice with PBS (0.5 mL, pH 7.4) and incubated with opsonized PS particles diluted at 1:10 with RPMI 1640 + 2% FCS for 1 h at 37 °C with 10% CO_2_. Particles were removed, and the cells were washed twice with PBS (0.5 mL, pH 7.4) and detached via incubation with 5 mM EDTA/PBS (0.5 mL, pH 7.4) for 10 min on ice. The cells were collected in FACS tubes, and the reaction was stopped with RPMI 1640 + 2% FCS (3× the volume of EDTA/PBS). The cells were centrifuged at 1700 rpm for 5 min and washed twice with PBS (0.5 mL, pH 7.4). Thereafter, the cells were re-suspended in PBS (0.1 mL, pH 7.4) and stained with a Zombie NIRTM Fixable Viability Kit (#423106, BioLegend) diluted at 1:500, and an anti-F4/80 AlexaFluor647 conjugated antibody (clone CI:A3-1, #MCA497A647, BioRad, Hercules, CA, USA) diluted at 1:200 in PBS (0.1 mL, pH 7.4) was added for 20 min at 4 °C in the dark. The cells were then washed with PBS (0.5 mL, pH 7.4) and centrifuged at 1700 rpm for 5 min. The supernatant was removed, and the cells were fixed in 0.1% PFA/PBS (0.2 mL, pH 7.4). Samples were acquired using a BD LSRFortessa™ Cell Analyzer equipped with 405 nm, 488 nm, 561 nm, and 640 nm laser lines and by using FACSDiva Software (BD Bioscience). GPR84 surface expression was quantified using manual gating in FlowJo™ v10.10 software.

### 4.7. Detection of Reactive Oxygen Species (ROS)

Isoluminol-based ROS assay: Undifferentiated human THP-1 cells were incubated with 50 ng/mL PMA for 3 h, seeded at a density of 100.000 cell/well (in RMPI 16,400 + 10% FCS) in a white 96-well plate (Greiner Bio-One), and incubated for 24 h at 37 °C with 5% CO_2_. On the day of the experiment, the cells were incubated either with the GPR84 antagonist GLPG1205 (HY-135303, MedChemExpress, Monmouth Junction, NJ, USA) at a concentration of 20 µM dissolved in DMSO or DMSO only for 20 min at room temperature (RT). The cells were washed with Krebs–Ringer solution before adding isoluminol and horseradish peroxidase (HRP). The cells were incubated for 5 min at RT before chemiluminescence baseline reading using an EnVision^®^ Xcite 2105 multimode plate reader (Perkin Elmer, Waltham, MA, USA). GPR84 agonists were added at different concentrations, and a kinetic chemiluminescence read was performed.

H2DCFDA-based ROS assay: Differentiated murine BMDMs (24-well plates: 0.5 × 10^6^ cells in 0.5 mL media) were stimulated ± 100 ng/mL LPS (InvivoGen) for 16 h at 37 °C with 5% CO_2_ and stained with 1 µM DHE DCFDA (Thermo Fisher Scientific) in pre-warmed RPMI 1640 without phenol red (Thermo Fisher Scientific) for 1 h at 37 °C with 10% CO_2._After washing with PBS (1 mL, pH 7.4), the cells were exposed to ZQ-16 for different time periods (incubation times and concentrations are stated in the respective figure legends). Cells treated with 12.5 μM H_2_O_2_ served as a positive control. The cells were rinsed twice with PBS, detached from the surface with 2 mM EDTA/PBS (0.5 mL, pH 7.4), collected in FACS tubes, centrifuged at 1700 rpm for 5 min, washed twice with PBS (0.5 mL, pH 7.4), and stained for apoptosis using an Annexin V- DY-634 PI Apoptosis Staining/Detection Kit (#ab21448, Abcam, Cambridge, UK) dissolved in ice-cold Annexin-binding buffer (10 mM HEPES, 140 mM NaCl, and 2.5 mM CaCl, pH 7.4) according to the manufacturer’s instructions. The cells were analyzed using an LSR II Fortessa cytometer equipped with 405 nm, 488 nm, 561 nm, and 640 nm laser lines and with FACSDiva Software (BD Bioscience). Populations of interest were defined by manual gating using FlowJo v10.10 (BD Bioscience).

### 4.8. Antibody-Dependent Cellular Phagocytosis (ADCP) Assay with BMDMs and MC38

MC38 cells were expanded in MC38 culture media (high-Glucose Dulbecco’s Modified Eagle Medium (DMEM) 4500 mg/L D-glucose supplemented with 10% FCS (#S181T-500, VWR, Radnor, PA, USA), 1 mM non-essential amino acids (NEAAs) (Thermo Fisher Scientific), and 1 mM Sodium Pyruvate (Thermo Fisher Scientific)) in 150 cm^2^ cell culture flasks (TPPs) at 37 °C with 10% CO_2_. On the day before the experiment, differentiated murine BMDMs were stimulated with 100 ng/mL LPS in RPMI 1640 + 2% FCS, while MC38 cells (approximately 75% confluence) were incubated with 10 µM CellTrackerTM Green CMFDA Dye (Thermo Fisher Scientific) for 16 h at 37 °C with 10% CO_2_. On the day of the experiment, MC38 cells were washed twice with PBS (10 mL, pH 7.4) and detached from the flasks using 5 mL 1× Trypsin/EDTA for 5 min at 37 °C with 10% CO_2_ (Thermo Fisher). The reaction was stopped by using MC38 culture media (3× the volume of Trypsin/EDTA), and MC38 cells were centrifuged at 1700 rpm for 5 min, counted, re-suspended in MC38 culture media at a ratio of 1 × 10^6^ cells/mL, and incubated with 10 µg/mL anti-CD47 antibody (#16-0471-81, Thermo Fisher Scientific) for 1 h at 37 °C with 10% CO_2_. Meanwhile, the LPS media was removed and BMDMs were incubated with 1 µM 6-OAU for 1 h at 37 °C with 10% CO_2_. MC38 cells were centrifuged at 1700 rpm for 5 min, washed twice with PBS (1 mL, pH 7.4), and re-suspended in RPMI 1640 + 2% FCS at a ratio of 1 × 10^6^ cells/mL. Subsequently, the 6-OAU media was removed from the BMDMs and the labeled MC38 cells with CD47 blockage were added at a ratio of 1:6 for 1 h at 37 °C with 10% CO_2_. Finally, the supernatant was removed, and the cells were washed twice with PBS (1 mL, pH 7.4) and detached using 5 mM EDTA/PBS (1 mL, pH 7.4) for 10 min on ice. The reaction was stopped with RPMI 1640 + 2% FCS (3× the volume of EDTA/PBS), and the cells were collected in FACS tubes, centrifuged at 1700 rpm for 5 min, and washed twice with PBS (1 mL, pH 7.4). The cells were re-suspended in PBS (1 mL, pH 7.4), stained with a Zombie NIRTM Fixable Viability Kit (#423106, BioLegend) diluted at 1:500 and an anti-F4/80 AlexaFluor647 conjugated antibody (clone CI:A3-1, #MCA497A647, BioRad) diluted at 1:200 in PBS (0.1 mL, pH 7.4) for 20 min at 4 °C in the dark, washed with PBS (0.5 mL, pH 7.4), and centrifuged at 1700 rpm for 5 min. The supernatant was removed and the cells were fixed in 0.1% PFA/PBS (0.2 mL, pH 7.4). Samples were acquired using a BD LSRFortessa™ Cell Analyzer equipped with 405 nm, 488 nm, 561 nm, and 640 nm laser lines and with FACSDiva Software (BD Bioscience). The proportion of double-positive macrophages/mean green fluorescence intensity of the macrophages was quantified by manual gating using FlowJo™ v10.10 software. Unlabeled BMDMs and MC38 cells served as staining controls. The expression of CD47 on the surface of MC38 cells was confirmed prior to the experiments (see Appendix A).

### 4.9. ADCP Assay with J774 and Raji Cells

ADCP assays using mouse J774 (#TIB-202TM ATCC) and human Raji (#CCL-86TM ATCC) cells were performed by Cellomatics Bioscience, Nottingham, UK. J774 cells were cultured in DMEM (Thermo Fisher Scientific) supplemented with 4 mM L-glutamine, 4500 g/L glucose, 1 mM sodium pyruvate, 1 mg/L sodium bicarbonate, and 10% FCS at 37 °C with 5% CO_2_. Cultures were maintained at the recommended sub-cultivation ratio of 1:3 to 1:6 and were split once the cultures reached ~80–90% confluence. Raji cells were cultured in RPMI-1640 (Gibco) supplemented with 2 mM L-glutamine, 10 mM HEPES, 1 mM sodium pyruvate, 4500 mg/L glucose, 1500 mg/L sodium bicarbonate, and 10% FCS at 37 °C with 5% CO_2_. Cultures were maintained at a density of 400,000 cells/mL and split once the density reached ~3 × 10^6^ cells/mL. Then, 48 h before co-culture initiation, J774 cells were harvested by scraping, counted, and seeded at 15,000 cells/well in a 96-well plate. After 24 h, the J774 macrophages were treated with 100 ng/mL LPS (#L4391-1MG, Sigma Aldrich) for 24 h prior to co-culture with the target Raji cells. Prior to co-culture, J774 effector cells were pre-treated with test compounds at 2× the desired concentration (0, 2, 20, 200, 2000, and 20,000 nM) for 1 h. J774 effector cells were labeled with calcein AM (#C1430 Thermo Fisher Scientific), while the Raji target cells were labeled with pHrodo Red labeling dye (#4649 Sartorius) for 1 h at 1 × 10^6^ cells/mL, according to the manufacture’s protocol. After labeling, Raji cells were pre-treated with different concentrations of magrolimab (#orb689306 Biorbyt) (0, 0.11, and 1 μg/mL) for 1 h. The labeled and treated Raji cells were seeded on top of the LPS-stimulated and compound-treated macrophages at an effector–target ratio of 1:1. The co-culture was maintained undisturbed in cell culture conditions and imaged at 0, 2, 4, and 6 h using the ImageXpress Pico System (Molecular Devices, San Jose, CA, USA). The effects of the magrolimab and test compound pre-treatment on macrophage–Raji cell co-cultures were evaluated by determining the proportion of target cells (red) that had been engulfed relative to the number of target cells seeded at the start of the co-culture. The overall intensity of the red fluorescent signal in the whole well was also determined. The relative red fluorescence intensity (RFU) and the percentage of target cells positive for phagocytosis were analyzed.

### 4.10. Subcutaneous MC38 Injection Model to Test Anti-Cancer Effect of 6-OAU In Vivo

The animal experiment was performed in the Zürich lab, conducted according to Swiss animal welfare legislation, and approved by the local veterinary office (Veterinary Office of the Canton Zürich; License ZH219/2022). Mice: 12-week-old female C57BL/6 wild-type (WT) mice purchased from Janvier Labs, Le Genest-Saint-Isle, France, were used for the experiment. Upon arrival at our facility, the mice were given 2 weeks of acclimatization time. The mice were kept in a specific pathogen-free (SPF) facility with chow and water ad libitum. Study design: The primary outcome of the study was the tumor volume measured in mm3. Randomization: Prior to the experiment, the mice were randomized using a random number generator. Blinding: Treatment administration was conducted blindly by using allocation groups to ensure that all animals in the experiment were handled, monitored, and treated the same way. Sample collection at the end of the experiment and sample analysis were conducted in a blinded manner. Statistics: This was a confirmatory experiment, so the null hypothesis was that there is no difference between the mean of all groups. The alternative hypothesis was that at least one of the paired comparisons would change the tumor size. Sample size calculations: The tumor cells were injected, and the compounds were administered via oral gavage for each mouse individually; thus, each mouse was considered an experimental unit. We employed a one-way Analysis of Variance (ANOVA) model, supplemented by pairwise multiple comparison procedures to discern between-group differences, with an adjusted alpha level of 0.025. Based on the estimated tumor volume standard deviation of 0.9 from previous experiments and a predetermined effect size of 2 mm, we conducted a power analysis using the Fpower1 function from the daewr R package (v1.2.11). After considering ethical considerations and resource constraints, we determined a sample size of seven animals per group. This sample size ensured 80% power to detect meaningful differences. Tumor cell injection: MC38 cells (isolated and provided from Prof. Dr. Lubor Borsig, University Zürich, Zürich, Switzerland) were expanded in MC38 culture media (high-Glucose Dulbecco’s Modified Eagle Medium (DMEM) 4500 mg/L D-glucose supplemented with 10% FCS (#S181T-500, VWR, Radnor, PA, USA), 1 mM non-essential amino acids (NEAAs) (Thermo Fisher Scientific), and 1 mM Sodium Pyruvate (Thermo Fisher Scientific)) in 150 cm2 cell culture flasks (TPPs) at 37 °C with 10% CO_2_. On the day of the experiment, MC38 cells (70–80% confluence) were washed twice with PBS (10 mL, pH 7.4) and detached from the flasks using 1× Trypsin/EDTA for 5 min at 37 °C with 10% CO_2_ (Thermo Fisher). The reaction was stopped with MC38 culture media (3× the volume of Trypsin/EDTA), and MC38 cells were centrifuged at 1700 rpm for 5 min, counted, and re-suspended in MC38 culture media at a density of 6 × 10^6^ cells/mL media. The cell suspension was then mixed with liquid growth-factor-reduced matrigel (#FAL354263, Corning, Corning, NY, USA) at 1:1 and kept on ice until injection. The mice were anesthetized using isoflurane inhalation anesthesia (5% isoflurane/1000 mL/min O_2_), and after surgical tolerance was reached, 300.000 MC38 and 100 µL of the cell suspension/matrigel mix were injected subcutaneously bilaterally into the flanks of each mouse using a 29G insulin syringe (Becton Dickinson AG, Franklin Lakes, NJ, USA). Groups were alternated during the tumor cell injection process. Compound preparation: 6-OAU was prepared fresh every day. Working solution of 10 mpk: To every 1 mg of 6-OAU powder, 50 µL DMSO was added, and the suspension was vortexed until the powder dissolved completely. Then, 150 µL Solutol (#42966-1KG, Sigma-Aldrich) was added, and the mixture was sonicated for 1 min. Finally, 800 µL 23% Captisol (#S4592-1G, Selleck Chemicals, Cologne, Germany, dissolved in H_2_O) was added (final concentration 6-OAU: 1 mg/mL). Working solution of 1 mpk: The 10 mpk solution was diluted at 1:10 with 5% DMSO, 15% Solutol, and 23% Captisol 80% H_2_O. A mixture of 5% DMSO, 15% Solutol, and 23% Captisol 80% H_2_O was used as a vehicle control. Compound administration: 6-OAU treatment was performed daily from day 6 until day 14. Then, 0, 1, or 10 mpk 6-OAU was administered via oral gavage (10 µL volume per g body weight). Monitoring: Health monitoring and body weight was recorded daily (parameters: appearance, behavior/activity, pain, scratching/licking of injection site, wounds/necrosis on tumors). The width and the length of the tumors were measured daily using an electronic caliper and the volume was calculated according to the following formula: = 4/3 × π × length/2 × width/2 × height/2. The mice were euthanized when the termination criteria were reached: wounds/necrosis on the tumors or tumor volume >1.5 cm^3^ (other termination criteria, including elevated appearance, behavior/activity scores, intense licking and scratching of the injection sites, signs of moderate pain, weight loss ≥ 10%, or elevated cumulative score were not observed in this study). Euthanasia: The mice were anesthetized by intraperitoneal injection of 120 mg/kg BW Ketamine (Dr. E. Gräub AG) + 16 mg/kg BW Xylazine (Streuli Tiergesundheit AG, Uznach, Switzerland) in PBS using a 29G insulin syringe. After surgical tolerance was reached, the mice were euthanized by complete exsanguination. Tumors were collected and weighed.

### 4.11. Subcutaneous MC38 Injection Model to Test Anti-Cancer Effect of ZQ-16 In Vivo

The ZQ-16 in vivo study was performed by Aragen Life Sciences Pvt. Ldt., Benaluru, India (Study Number 019-23-PH). Ethical considerations: The animal experiment was approved by the Institutional Animal Ethics Committee (IAEC) in accordance with the requirement of the Committee for the Purpose of Control and Supervision of Experiments on Animals (CPCSEA), India. Mice: 6–7-week-old female C57BL/6 wild-type (WT) mice purchased from Taconic (Vivo Biotech Pvt. Ltd., Telangana, India) were used for the experiment. The mice were kept in an SPF facility with chow and water ad libitum. Study design: The primary outcome of the study was the tumor volume measured in mm^3^. Randomization: The mice were randomized on day 7 based on the tumor volume to equalize the average tumor volume between groups. Blinding: Treatment administration was conducted blindly by using allocation groups to ensure that all animals in the experiment were handled, monitored, and treated the same way. Sample collection at the end of the experiment and sample analysis were conducted in a blinded manner. Statistics: This was a confirmatory experiment, and hence, the null hypothesis stated no difference between the mean of all groups. The alternative hypothesis asserted that at least one of the paired comparisons would change the tumor size. Sample size calculations: The tumor cells were injected, and the compounds were administered via intraperitoneal injection or oral gavage for each mouse individually, considering each mouse as an experimental unit. We employed a one-way Analysis of Variance (ANOVA) model, supplemented by pairwise multiple comparison procedures to discern between-group differences, with an adjusted alpha level of 0.025. Based on the estimated tumor volume standard deviation of 0.9 from previous experiments and a predetermined effect size of 2 mm, we conducted a power analysis using the Fpower1 function from the daewr R package (v1.2.11). After considering ethical considerations and resource constraints, we determined a sample size of seven animals per group. This sample size ensured 80% power to detect meaningful differences. The experiment was performed with n = 10 mice per group. Tumor cell injection: MC38 cells (#ENH204-FP, Kerafast) were expanded in DMEM media supplemented with 10% FCS at 37 °C with 10% CO_2_. Then, 1 × 10^6^ MC38 in 100 µL media were injected subcutaneously into each flank. Compound preparation: IP: 3 mg of ZQ-16 was weighed and 0.131 mL of DMSO was added, and then 14 aliquots of 0.005 mL for 14 days of dosing were prepared. Working solution: For 2.5 mL of IP formulation, 0.005 mL of aliquot stock solution was diluted with 2.495 mL saline. Saline with 0.5% DMSO was used as a vehicle control. PO: 2.5 mg of the test compound was weighed and dissolved in 1.0 mL of PEG-400 by means of vortexing for 1–2 min and sonicating for 1–2 min. Then, 0.750 mL PG was added, vortexed for 1–2 min, and sonicated for 1–2 min. Finally, 0.750 mL water was added, and the solution was vortexed for 1–2 min to obtain a uniform mixture. Then, 40% PEG400 and 30% PG in H_2_O were used as a vehicle control. All of the formulations were prepared fresh before dosing on each day. Compound administration: ZQ-16 treatment was performed daily from day 7 until day 21. IP: 0 or 0.457 mpk 6-OAU was administered via intraperitoneal injection (10 µL volume per g body weight). PO: 0 or 10 mpk 6-OAU was administered via oral gavage (10 µL volume per g body weight). Monitoring: Everyday general clinical signs and body weight were monitored. Tumor volume was measured every second day with digital vernier calipers by using the formula (LXW^2^)/2, where L is the largest diameter and W is the smallest diameter. Euthanasia: Animals were euthanized 6 h and 24 h after the last treatment. Blood was collected into serum tubes under isoflurane anesthesia, allowed to coagulate for half an hour, spun down, and serum-collected into fresh tubes. Tumors were weighed and collected from all groups and subsequently snap-frozen.

### 4.12. Statistical Analysis

Statistical analyses were performed using one- or two-way ANOVA with Tukey’s multiple comparison or Dunnett’s multiple comparison post hoc test. The results are expressed as mean ± SEM, and significance was set as *p* < 0.1. Graph Pad Prism 9 was used to create the graphs and for conducting the statistical tests.

## Figures and Tables

**Figure 1 ijms-26-00490-f001:**
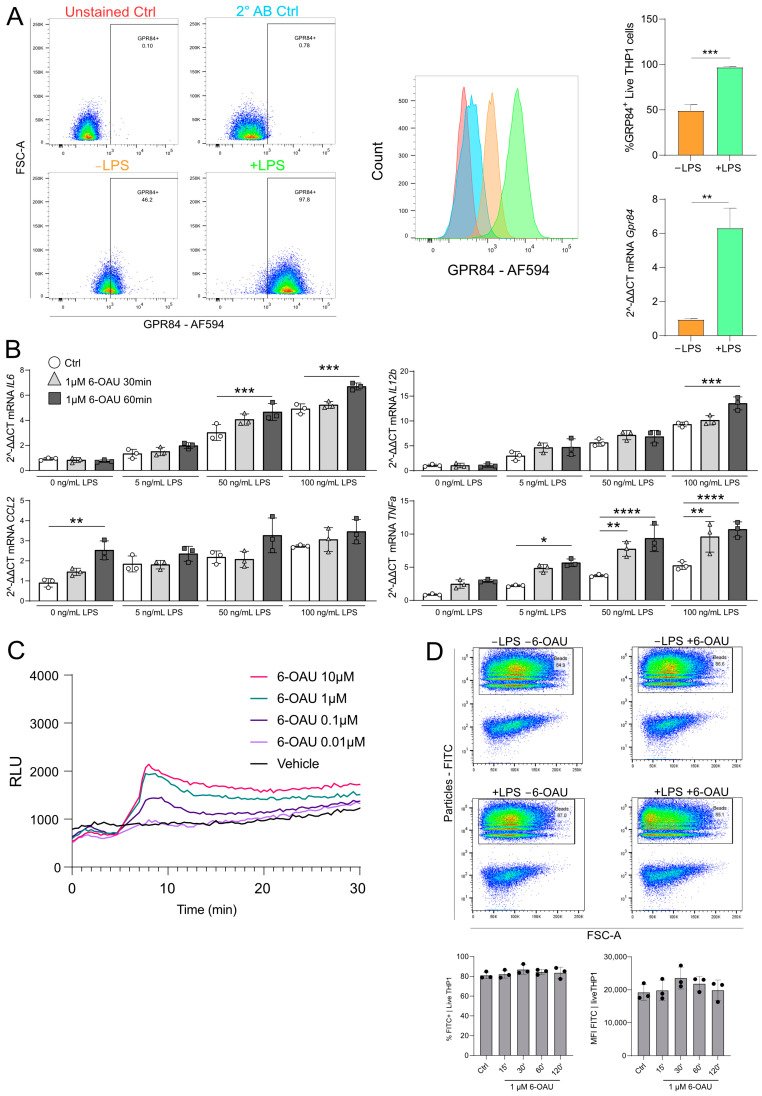
In THP-1 cells, LPS increased GPR84 expression, and 6-OAU enhanced pro-inflammatory cytokine transcription. (**A**) Flow cytometry of GPR84 surface and mRNA expression after LPS stimulation (100 ng/mL, 16 h). Bar charts quantify expression levels. (**B**) mRNA expression of GPR84 targets after LPS (0–100 ng/mL) and 6-OAU (1 µM) stimulation. (**C**) ROS production kinetics in response to 6-OAU (0.01–10 µM) treatment. (**D**) Phagocytosis of opsonized beads with LPS (100 ng/mL, 16 h) and 6-OAU (1 µM, 15–120 min) treatment. Statistical analysis: (**A**) Student’s *t*-test; (**B**–**D**) one-way ANOVA and Tukey’s test, with * *p* ≤ 0.05, ** *p* ≤ 0.01, *** *p* ≤ 0.001, and **** *p* ≤ 0.0001. n = 3.

**Figure 2 ijms-26-00490-f002:**
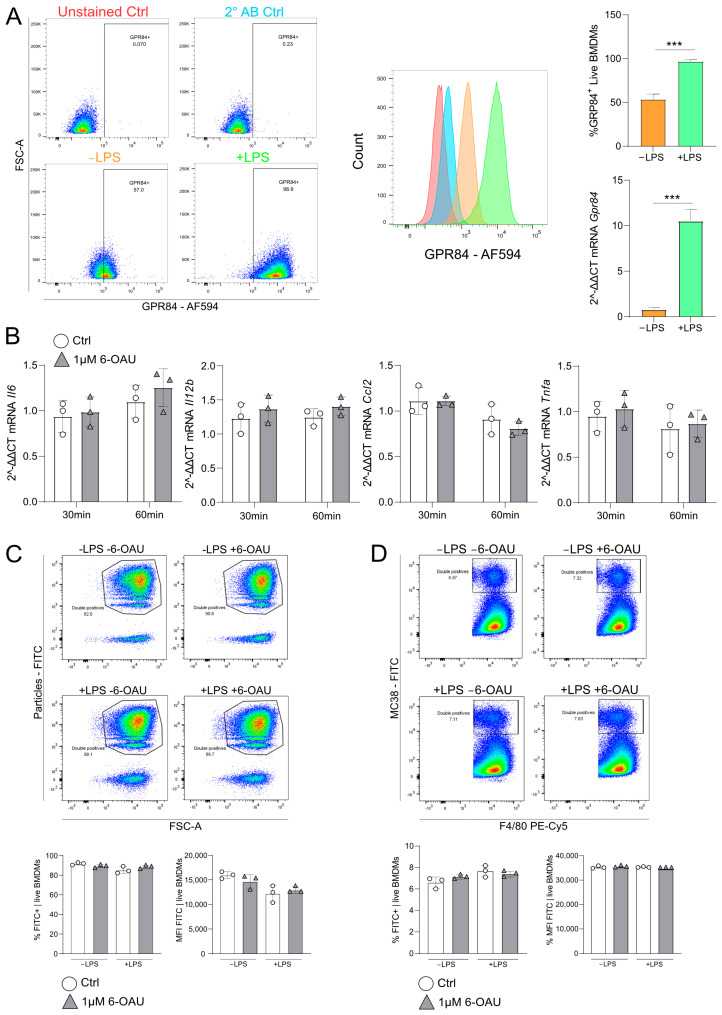
In murine BMDMs, LPS increased GPR84 expression, but 6-OAU did not enhance pro-inflammatory signaling or phagocytosis. (**A**) Flow cytometry of GPR84 surface and mRNA expression after LPS (100 ng/mL, 16 h) stimulation. Bar charts quantify expression. (**B**) mRNA expression of GPR84 targets after LPS (100 ng/mL, 16 h) and 6-OAU (1 µM, 30 or 60 min) stimulation. (**C**) Phagocytosis of opsonized beads after LPS (100 ng/mL, 16 h) and 6-OAU (1 µM, 60 min) treatment. (**D**) Phagocytosis of CD47-blocked MC38 cells after LPS (100 ng/mL, 16 h) and 6-OAU (1 µM, 60 min) treatment. Statistical analysis: (**A**) Student’s *t*-test; (**B**–**D**) one-way ANOVA and Tukey’s test, with *** *p* ≤ 0.001. n = 3.

**Figure 3 ijms-26-00490-f003:**
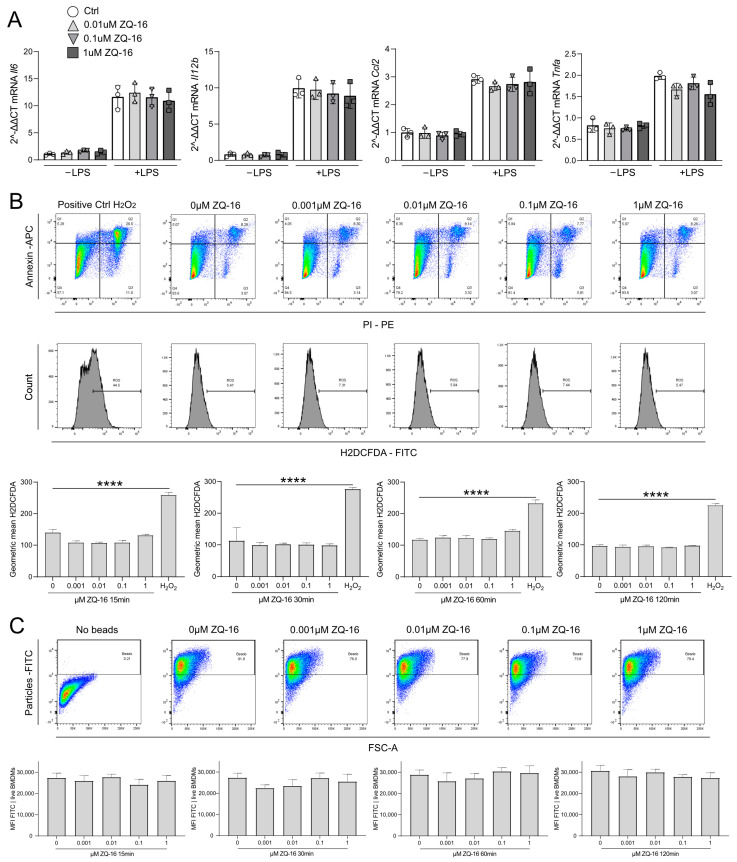
In murine BMDMs, ZQ-16 did not enhance pro-inflammatory cytokine expression, phagocytosis, or ROS production. (**A**) mRNA expression of GPR84 downstream molecules after LPS (100 ng/mL, 16 h) and ZQ-16 (0.01–1 µM, 60 min) stimulation. (**B**) ROS levels after LPS (100 ng/mL, 16 h) and ZQ-16 (0.01–1 µM, 60 min) treatment analyzed by flow cytometry. (**C**) Phagocytosis of opsonized beads after LPS (100 ng/mL, 16 h) and ZQ-16 (0.001–1 µM, 60 min) treatment. Statistical analysis: one-way ANOVA and Tukey’s test, with **** *p* ≤ 0.0001. n = 3.

**Figure 4 ijms-26-00490-f004:**
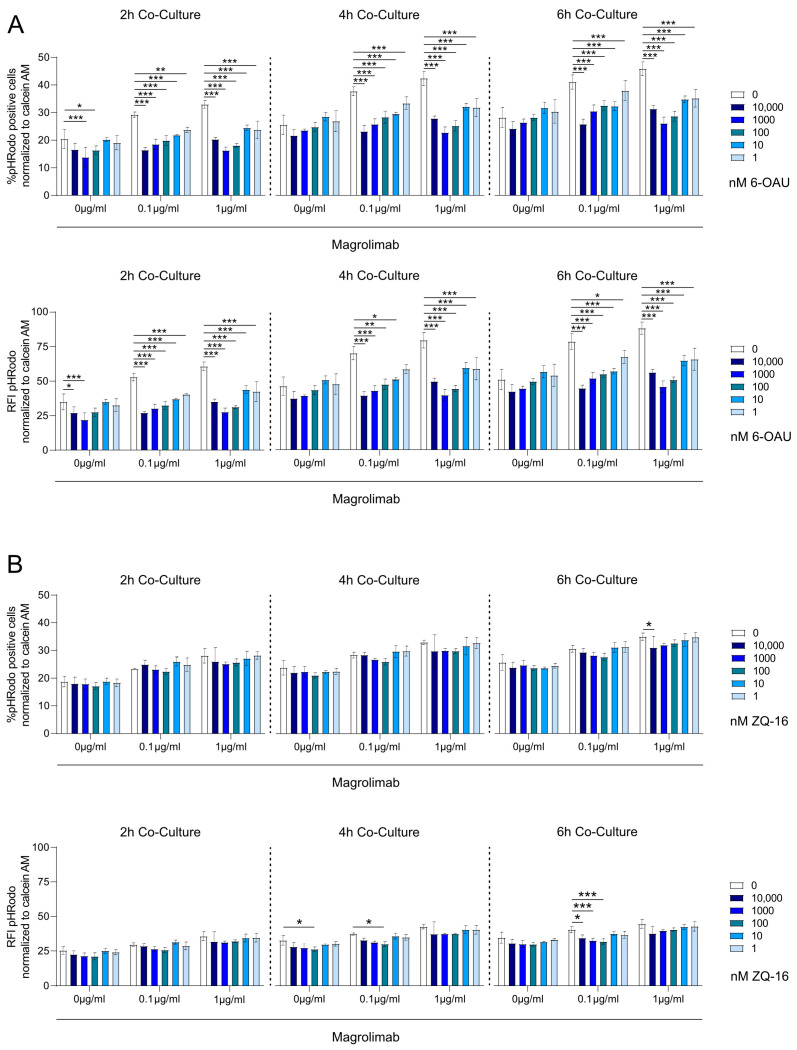
GPR84 agonists did not enhance J774-mediated phagocytosis of Raji cells. J774 macrophages, pre-treated with (**A**) 6-OAU or (**B**) ZQ-16, were incubated with LPS and co-cultured with pHrodo Red-labeled Raji cells and magrolimab for 6 h. Graphs show phagocytosis percentage and relative fluorescence intensity (RFI) of Raji cells at 2, 4, and 6 h. Data are normalized to macrophage numbers per well (n = 3 ± SEM). Statistical analysis: two-way ANOVA and Dunnett’s test, with * *p* < 0.05, ** *p* ≤ 0.01, and *** *p* ≤ 0.001.

**Figure 5 ijms-26-00490-f005:**
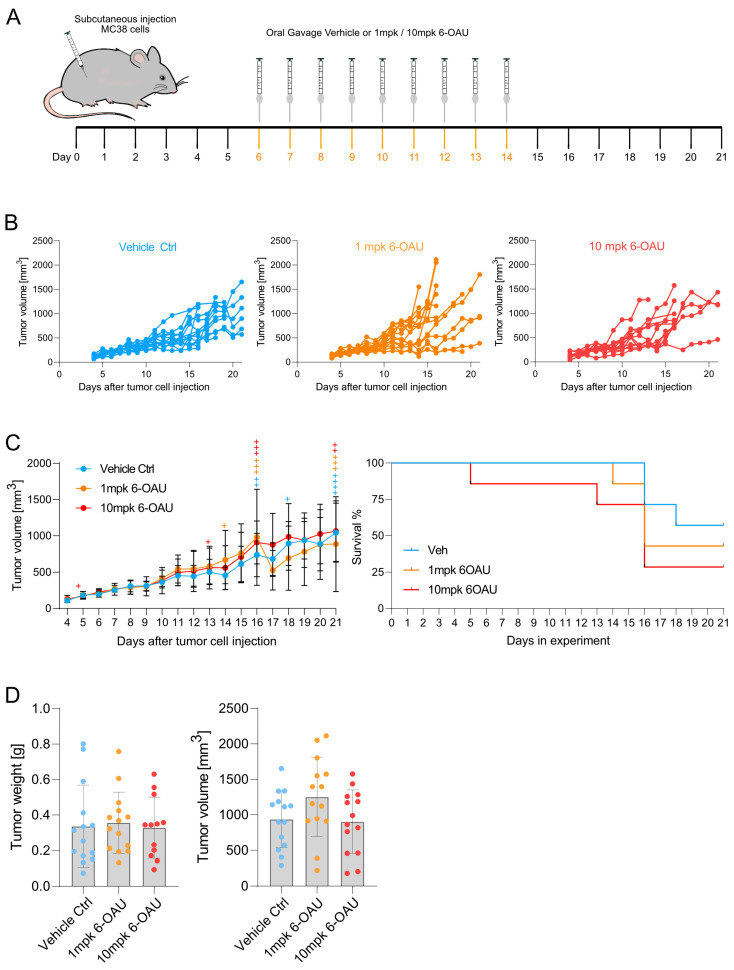
Oral 6-OAU administration showed no anti-cancer effects in the subcutaneous MC38 mouse model. C57BL/6 mice were injected with MC38 cells and treated with the vehicle or 6-OAU (1 or 10 mpk) via oral gavage. (**A**) The experimental design. (**B**) The tumor growth curve for each group. (**C**) Left: the average tumor growth and survival curve; right: the Kaplan–Meier curve. + indicates the removal of individual animals from experiments upon meeting termination criteria. (**D**) Tumor volume and weight at termination. Statistical analysis: (**B**,**C**) two-way ANOVA with Dunnett’s test; (**D**) one-way ANOVA with Tukey’s test.

**Figure 6 ijms-26-00490-f006:**
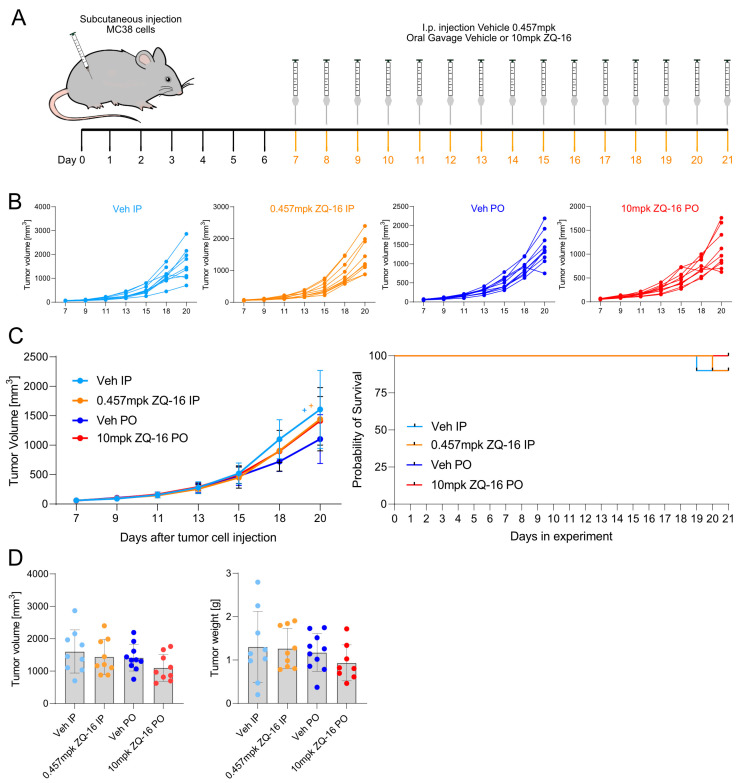
Oral and intraperitoneal ZQ-16 administration showed no anti-cancer effects in the subcutaneous MC38 mouse model. C57BL/6 mice were injected with MC38 cells and treated with the vehicle or ZQ-16 (0.457 mpk IP or 10 mpk PO). (**A**) The experimental design. (**B**) The tumor growth curve for each group. (**C**) Left: the average tumor growth and survival curve; right: the Kaplan–Meier curve. + indicates the removal of individual animals from experiments upon meeting termination criteria. (**D**) Tumor volume and weight at termination. Statistical analysis: (**B**,**C**) two-way ANOVA with Dunnett’s test; (**D**) one-way ANOVA with Tukey’s test.

## Data Availability

The original contributions presented in this study are included in the article/Appendix A. Further inquiries can be directed to the corresponding author(s).

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
