# Peer review of "Synthetic GPR84 Agonists in Colorectal Cancer: Effective in THP-1 Cells but Ineffective in BMDMs and MC38 Mouse Tumor Models"

_ijms, 2025, doi:10.3390/ijms26020490_

Round 1

Reviewer 1 Report

Comments and Suggestions for Authors

The negative or inconclusive results of the studies are important for identifying new approaches in research. However, the authors' hypothesis was not supported by the results, which provides valuable insights for the future direction of this research.

Considering animal welfare and ethical aspects, the authors should have stopped the study after the negative outcomes from the in-vitro experiments. Nonetheless, this provides further evidence that the two G-protein receptors are not viable targets for this new approach. With a few minor corrections and additions, this manuscript is suitable for publication.

1. The graphical abstract or a figure illustrating the mechanism of action for the activation of the G protein-coupled receptor 84 (GPR84), associated with pro-inflammatory macrophage polarization, serves as an effective introduction and engages the reader's interest.

2. For Figures 1, 2, and 3, it is recommended to distinguish the bars from their counterparts using symbols, especially if they represent any significant differences, as mentioned in the figure legends. Additionally, in line 158 of the figure legend, the notation “n=3.2.2” is unclear and should be clarified.

3. There are also a few typos in the manuscript that need to be addressed. For example, in line 652, “Aragen Life Sciences Pvt. Ldt., Benaluru, India” should be corrected to “Aragen Life Sciences Pvt. Ltd., Bengaluru, India.”

Author Response

Comment 1: The graphical abstract or a figure illustrating the mechanism of action for the activation of the G protein-coupled receptor 84 (GPR84), associated with pro-inflammatory macrophage polarization, serves as an effective introduction and engages the reader's interest.

Response 1: We thank the reviewer for this valuable feedback. To address this, we have included a graphical abstract in the manuscript that summarizes the key findings of the study.

Comment 2: For Figures 1, 2, and 3, it is recommended to distinguish the bars from their counterparts using symbols, especially if they represent any significant differences, as mentioned in the figure legends. Additionally, in line 158 of the figure legend, the notation “n=3.2.2” is unclear and should be clarified.

Response 2: We appreciate the reviewer’s input. The figures have been adjusted as suggested, and the error in line 158 has been corrected.

Comment 3: There are also a few typos in the manuscript that need to be addressed. For example, in line 652, “Aragen Life Sciences Pvt. Ldt., Benaluru, India” should be corrected to “Aragen Life Sciences Pvt. Ltd., Bengaluru, India.

Response 3: We thank the reviewer for highlighting this mistake. Additional typos will be addressed during the proofreading process.

Reviewer 2 Report

Comments and Suggestions for Authors

This paper investigates the potential of synthetic GPR84 agonists (6-OAU and ZQ-16) as therapeutic agents for colorectal cancer (CRC). The study evaluated their effects on macrophage activation and anti-cancer efficacy. While the agonists induced pro-inflammatory responses and reactive oxygen species (ROS) production in human THP-1 macrophages, they failed to show similar effects in murine bone marrow-derived macrophages (BMDMs). Furthermore, neither agonist improved macrophage-mediated phagocytosis nor inhibited tumor growth in the MC38 mouse model of CRC. While this study provides a broad overview of the positive and negative effects of GPR84 agonists on macrophages and tumor growth, it falls short of meeting the rigorous standards of this journal due to limitations in experimental design, reproducibility, and translational relevance.

-       The authors should carefully review the presentation of data in this manuscript. For example, Figure 1 and Figure 2 appear to present overlapping data, which raises concerns about redundancy and the clarity of the results being communicated.

-       The introduction section is overly long and tedious. This extensive background makes it challenging for readers to grasp the core goals and significance of the research. A more concise and focused introduction would enhance the manuscript's clarity and impact.

-       As described in the Introduction and Discussion sections, the experimental design of this study overlaps significantly with several previous studies. While the results differ from those earlier findings, the novelty of this research is not adequately demonstrated, raising concerns about its contribution to advancing the field.

-       While the background highlights GPR84 as a potentially significant target in colorectal cancer, the results of this study do not substantiate this claim. The authors should consider exploring diverse GPR84-targeting strategies and conducting detailed functional studies to elucidate the precise mechanisms of GPR84 in colorectal cancer.

Author Response

Comment 1: The authors should carefully review the presentation of data in this manuscript. For example, Figure 1 and Figure 2 appear to present overlapping data, which raises concerns about redundancy and the clarity of the results being communicated.

Response 1: We appreciate the reviewer’s feedback and have carefully reviewed the presentation of our data. While Figures 1 and 2 share similar experimental designs to evaluate the effects of the GPR84 agonist 6-OAU, the datasets are distinct and not redundant. Figure 1 focuses on human THP-1 macrophages, examining the surface expression of GPR84, the impact of 6-OAU on pro-inflammatory cytokine transcription, ROS production, and phagocytosis in this immortalized human cell line. In contrast, Figure 2 investigates primary murine bone marrow-derived macrophages (BMDMs), addressing the same parameters but within a physiologically relevant primary cell model, which is particularly important given the later use of mouse models in this study. Notably, the results differ between the two systems, as 6-OAU induced pro-inflammatory signaling and ROS production in THP-1 cells but failed to do so in murine BMDMs. This comparison highlights critical differences between immortalized human cell lines and primary murine macrophages, underscoring the translational challenges in GPR84-targeted therapies. Thus, the data are complementary rather than redundant, contributing to the clarity and depth of the study's findings.

Comment 2: The introduction section is overly long and tedious. This extensive background makes it challenging for readers to grasp the core goals and significance of the research. A more concise and focused introduction would enhance the manuscript's clarity and impact.

Response 2: We appreciate the reviewer’s criticism regarding the length and structure of the introduction. We revised this section to make it more concise and focused. Specifically, we streamlined the background information, limiting it to the most relevant details necessary to contextualize the study (lines 34-67). We also clearly articulated the core goals and significance of the research to ensure that readers can quickly grasp the purpose and importance of our work (lines 150-158).

Comment 3: As described in the Introduction and Discussion sections, the experimental design of this study overlaps significantly with several previous studies. While the results differ from those earlier findings, the novelty of this research is not adequately demonstrated, raising concerns about its contribution to advancing the field.

Response 3: We thank the reviewer for this insightful comment. While our findings may seem negative regarding the therapeutic efficacy of synthetic GPR84 agonists for CRC, negative data is a valuable and often underappreciated contribution to scientific progress. The inability to replicate prior results or achieve expected effects helps refine experimental designs, agonist selection, dosing regimens, and model systems. Our study highlights the need to explore species- and cell-type-specific differences in GPR84 responses and the influence of the microbiome on therapeutic outcomes. By publishing these findings, we aim to prevent premature advancement of GPR84 agonists into clinical trials without addressing these crucial variables. Although our work overlaps with previous studies, its novelty lies in exploring discrepancies across species and cell types that could influence GPR84 activation, guiding future research. In this way, negative data provides critical information that steers future investigations, ensuring more effective cancer therapies. This aspect is now addressed in the conclusion (lines 418-430).

Comment 4: While the background highlights GPR84 as a potentially significant target in colorectal cancer, the results of this study do not substantiate this claim. The authors should consider exploring diverse GPR84-targeting strategies and conducting detailed functional studies to elucidate the precise mechanisms of GPR84 in colorectal cancer.

Response 4: We thank the reviewer for this valuable comment. While our study does not fully substantiate GPR84 as a significant therapeutic target in CRC, it provides important insights into the complexities and potential limitations of targeting GPR84 for treatment. As suggested, exploring diverse GPR84-targeting strategies, including biased agonism, alternative receptor pathways, and the microbiome’s role in modulating GPR84 activity, is crucial for advancing our understanding of its therapeutic potential. Although further functional studies to elucidate GPR84’s precise mechanisms in CRC are necessary, these are beyond the scope of our current work. We hope our study will inspire further investigations into species- and cell-type-specific effects of GPR84 activation, optimized dosing regimens, and more clinically relevant experimental models. These future studies will help define GPR84’s role in CRC and clarify its potential as a therapeutic target. This aspect is now covered in the conclusion (lines 418-430).

Round 2

Reviewer 2 Report

Comments and Suggestions for Authors

The authors responded appropriately to comments.